# The Impact of Climatological Factors on the Incidence of Cutaneous Leishmaniasis (CL) in Colombian Municipalities from 2017 to 2019

**DOI:** 10.3390/pathogens13060462

**Published:** 2024-05-30

**Authors:** Daniel Muñoz Morales, Fernanda Suarez Daza, Oliva Franco Betancur, Darly Martinez Guevara, Yamil Liscano

**Affiliations:** 1Facultad de Salud, Universidad de Caldas, Manizales 170004, Colombia; fernandasuarezdaza@gmail.com (F.S.D.); olifrancob@gmail.com (O.F.B.); 2Grupo de Investigación en Salud Integral (GISI), Departamento Facultad de Salud, Universidad Santiago de Cali, Cali 760035, Colombia

**Keywords:** leishmania, leishmaniasis, correlation coefficient, correlation analysis, climate, climatic variables, Colombia, Spearman

## Abstract

Cutaneous leishmaniasis (CL) is a zoonotic disease caused by protozoa of the Leishmania genus, transmitted by vectors from the Phlebotominae subfamily. The interaction between the vector, reservoir, and parasite is susceptible to climate change. This study explores how temperature and rainfall influenced the incidence of CL in 15 Colombian municipalities between 2017 and 2019. Epidemiological data were obtained from Colombia’s Instituto Nacional de Salud, while climatological data came from the Instituto de Hidrología, Meteorología y Estudios Ambientales. Using Spearman’s rank correlation coefficient, we examined the relationships between monthly climatic variables and the cumulative incidence of CL, considering various lag times. The data were further analyzed using Locally Weighted Scatterplot Smoothing (LOWESS). Our findings reveal both significant positive and negative correlations, depending on locality and climate variables. LOWESS analysis indicates that while rainfall-related incidence remains stable, temperature impacts incidence in a parabolic trend. This study underscores the significant yet complex influence of climatic factors on CL incidence. The insights gained could aid public health efforts by improving predictive models and crafting targeted interventions to mitigate the disease’s impact, particularly in regions vulnerable to climate variability.

## 1. Introduction

Leishmaniasis is a zoonotic infectious disease caused by at least 20 protozoan species from the *Leishmania* genus, transmitted among mammals through the bite of female Diptera from the Psychodidae family [1]. Recognized by the World Health Organization (WHO) as a neglected tropical disease, it is endemic in the tropical and subtropical regions of the world [2]. The disease is especially prevalent in developing countries across regions such as Asia, including China and Southeast Asia, the Middle East, North Africa, Southern Europe, and, in the new world, extends from Mexico through Central America and parts of South America [2,3]. Brazil and Colombia are America’s countries reporting the highest number of cutaneous leishmaniasis (CL) cases [4], with Colombia documenting 5929 leishmaniasis cases in 2019, of which 5839 were CL [5].

The disease manifests in three clinical forms: CL, mucocutaneous leishmaniasis (MCL), and visceral leishmaniasis (VL), with CL being the most common yet least severe form [6]. The vector acquires the protozoan while feeding on the reservoir host; the protozoan then develops in the arthropod’s intestine until it reaches its infectious form [1], enabling the vector to transmit the parasite to humans or another reservoir to perpetuate the cycle through a bite. This intricate biological transmission mechanism underscores the critical influence of environmental factors, especially climate change, in determining the distribution and prevalence of vector-borne diseases such as leishmaniasis [1].

Leishmaniasis exerts a profound socioeconomic impact on communities, especially in regions where it is endemic [7,8,9,10]. The disease extends its influence beyond mere health implications, substantially affecting the economic stability and productivity of the affected individuals and communities. Treatment costs, often disproportionate to local incomes, can drive families into severe financial distress, compelling them to sell assets or secure loans at high interest rates to cover medical expenses. Additionally, the productivity loss due to illness or caregiving for afflicted family members places further strain on household resources. These economic challenges are further exacerbated by the impacts on local health systems, which in resource-poor settings may struggle with the costs of diagnosis, treatment, and ongoing management of leishmaniasis outbreaks [7,8,9,10].

Valero et al., 2021 [11] emphasize how changes in land use, including increased urbanization and deforestation, create new habitats for vectors and increase human exposure to diseases. This study highlights the interconnectedness of environmental changes and socioeconomic factors, underscoring the need for integrated public health strategies. These strategies should address not only the immediate health impacts but also the environmental and socioeconomic determinants of leishmaniasis.

The impact of climate on the behavior of vector-borne diseases has shown a growing concern in recent decades, with warnings about the potential effects of climate change on these diseases, including the expansion of vectors into new geographical and altitudinal zones [12,13,14,15]. Arthropod vectors of leishmaniasis and other vector-borne infections are sensitive to climatic conditions [15,16,17], with temperature, humidity, rainfall, and other factors influencing vectors and reservoirs by altering their survival, distribution, and population density [18].

Numerous studies have explored the relationship between leishmaniasis and environmental factors like altitude, latitude, temperature, humidity, precipitation, vegetation, land use, and the ENSO phenomenon. In Eurasia and Africa, research shows variable climate effects, with nonlinear relationships between rainfall, humidity, temperature, and disease incidence, alongside seasonal fluctuations. Some studies find an inverse relationship between air temperature and CL incidence, while others report increased incidence with higher temperatures. In Sri Lanka, humidity and maximum temperature positively correlate with leishmaniasis, whereas other regions show negative correlations with temperature. In Central America, temperature and rainfall are linked to high disease prevalence, and in the Amazon, both positive and negative relationships between climatic factors and CL incidence are observed [11,12,19,20,21,22,23,24,25,26].

Previous studies in Colombia have explored the temporal and spatial distribution of CL, the distribution of potential vectors, and their relationships with environmental factors to identify transmission risks using spatial epidemiology studies and ecological models [27,28,29,30]. Other studies sought an association between the incidence of CL and climatic variation given by ENSO, finding an influence of this cyclical meteorological phenomenon on the occurrence of the disease [12,31,32]. Despite research on other environmental factors and their influence on leishmaniasis incidence, there remains no consensus on the nature of the relationship between temperature, rainfall, and the incidence of CL [24,33,34].

Understanding the environmental factors that influence the behavior of leishmaniasis can facilitate the creation of predictive models or the application of public health measures to mitigate the disease’s impact on the population [18,25,35]. Given the multiple species of phlebotomine sandflies that act as effective vectors and the different species of Leishmania involved with distinct geographical distributions, the impact of climatic variables on each of these elements can differ, and consequently, the influence of climate on the variation of incidence patterns can also vary by region. This study aims to determine the impact of temperature and rainfall fluctuations on the incidence of CL in 15 municipalities of Colombia between 2017 and 2019 through a lagged correlation analysis, taking into account the average incubation period of the disease. This statistical analysis technique allows for the correlation of quantitative variables with lag times, explained by the time elapsed from the vector bite to the manifestation of the disease, offering insights into how the climatological state at a given time can stimulate the activity of the vector positively or negatively [17].

## 2. Materials and Methods

### 2.1. Epidemiological Data

Reported cases of CL, confirmed by laboratory tests from January 2017 to December 2019, were collected from 15 municipalities in Colombia through open-access databases published by the Instituto Nacional de Salud from Colombia (https://portalsivigila.ins.gov.co/Paginas/Buscador.aspx, accessed on 18 September 2023, Bogotá, Colombia). These databases included anonymized sociodemographic information for each case. Cumulative incidences were calculated at monthly intervals for each of the 36 months within the study period. To perform this calculation, the total number of new cases reported in a given month served as the numerator, while the denominator was derived from population data provided by the Departamento Administrativo Nacional de Estadísticas (https://www.dane.gov.co/index.php/estadisticas-por-tema/demografia-y-poblacion/proyecciones-de-poblacion, accessed on 18 September 2023, Bogotá, Colombia). These municipalities were chosen because they report cases of leishmaniasis continuously throughout the year, have variable climates, and have a population of less than 500,000 inhabitants. Additionally, there have been some of the most endemic localities year by year in the country.

### 2.2. Climatological Data

Temperature and rainfall data were obtained in monthly intervals from January 2017 to December 2019 from the Instituto de Hidrología, Meteorología y Estudios Ambientales (IDEAM) (https://dhime.ideam.gov.co/webgis/home/, accessed on 18 September 2023, Bogotá, Colombia), which has historical and public access data from various meteorological stations in the country. The data included total monthly rainfall (TMR), average monthly temperature (T_AVER), absolute maximum monthly temperature (T_ABS_MAX), absolute minimum monthly temperature (T_ABS_MIN), average maximum monthly temperature (T_AVER_MAX), average minimum monthly temperature (T_AVER_MIN), maximum temperature difference (T_DIF), and average temperature difference (T_AVER_DIF) at monthly intervals for each municipality of interest. Rainfall data were obtained for the 15 municipalities, but temperature data were gathered for only five of them due to a lack of data availability.

### 2.3. Statistical Analysis

Correlation analysis: To analyze the relationship of climatic variables (rainfall and temperature) with the cumulative incidence of CL, a Spearman’s Rank correlation [36] was performed with four different lag times as follows: each climate variable was correlated with its respective cumulative incidence with one, two, three, and four-month-long lag, performing a total of 4 correlation analyses for the total monthly rainfall variable for each municipality (analyzing a total of 15 municipalities) and 28 correlation analyses for the temperature variables for each municipality (analyzing a total of 5 municipalities), obtaining a correlation coefficient, the *p*-value as hypothesis testing with a significance on 0.05 and a 95% confidence interval (CI). The lag times were chosen based on the average incubation period of the disease reported in the literature [6,37,38].

### 2.4. General Analysis of Incidence in Relation to Climate

For global or general analysis, a LOWESS (Locally Weighted Scatterplot Smoothing) was applied to the two variables that showed the highest number of statistically significant correlations with the aim of generating an approximation of the possible relationships between climatological and epidemiological variables through a statistical “smoothing” by visually determining the patterns in a scatter plot. The described statistical approach was previously used by Peña-García et al. in the study of dengue [17].

For the statistical analyses, IBM SPSS Statistics software version 25 (https://www.ibm.com/support/pages/downloading-ibm-spss-statistics-25 SPSS License from Universidad de Caldas accessed on 18 September 2023, Armonk, NY, USA) was used, graphs and tables were created in Excel and SPSS, and finally, QGIS version 3.36.2 (https://qgis.org/es/site/forusers/download.html, accessed on 18 September 2023, Zurich, Suiza) was used as a geographic information system to perform georeferenced mapping of the correlation results on the map of Colombia. For this, vector layers obtained from the Departamento Administrativo Nacional de Estadística were used (https://geoportal.dane.gov.co/servicios/descarga-y-metadatos/descarga-mgn-marco-geoestadistico-nacional/, accessed on 18 September 2023, Bogotá, Colombia).

### 2.5. Ethical Considerations

During the development of this study, no interventions were made in the demographic and physiological variables of the participants. Therefore, this research work is considered minimal risk according to Resolution No. 8430 of 1993 of Colombian legislation and the Declaration of Helsinki. The study was approved by the bioethics committee of the Universidad de Caldas in ACT No. 011 of 2022.

## 3. Results

### 3.1. Sociodemographic Description

This study covers 6141 cases reported in the 15 selected municipalities, accounting for 32.2% of the total cases reported in Colombia over these three years. The median age of the cases was 24 years, with an interquartile range (IQR) of 19 to 31 years. Most of the cases were males, particularly within the 20- to 29-year age group, which accounts for 43.93% of all cases (Table 1). Notably, the 10- to 19-year age group also showed a substantial number of cases, constituting 18.55% of the total. There was a noticeable decline in the annual case numbers, with 36% in 2017, 33.5% in 2018, and 30.5% in 2019. The majority of cases were reported in rural areas, comprising 86.9% of the total. Occupational data reveal that military personnel and farmers/forest workers were disproportionately affected, making up 41.99% and 20.42% of the cases, respectively. There was a notably higher incidence among occupations associated with outdoor work, especially those exposed to wooded or jungle environments. Additionally, 65.05% of the cases were covered under the subsidized social security regime, indicating a prevalence of leishmaniasis among economically vulnerable populations.

The results of Table 2, covering 15 municipalities over the years 2017 to 2019, reveal critical insights into the disease’s distribution across various geographical and altitudinal contexts in Colombia. Notably, San José del Guaviare and Tumaco have high case numbers, with 1018 and 1934 cases, respectively. These municipalities are situated at relatively low altitudes (189 M.A.S.L. and 3 M.A.S.L.), indicating a high presence of leishmaniasis in these areas, which may be related to coastal or near-river environments. The distribution of cases also shows significant regional variability. For example, Antioquia and Bolívar report considerable case numbers across their municipalities, pointing to potential regional hotspots influenced by local environmental and socio-economic factors.

### 3.2. Correlation of Climatological Factors with Leishmaniasis Incidence across Colombian Municipalities

#### 3.2.1. Spearman’s Correlation Analysis of Temperature and Rainfall Variables

Figure 1 shows Spearman’s correlation coefficients for each municipality, examining the relationship between various climatological variables (TMR, T_AVER, T_ABS_MAX, T_ABS_MIN, T_AVER_MAX, T_AVER_MIN, T_DIF, and T_AVER_DIF) and the cumulative incidence over four lag times. It condenses the data to illustrate the impacts spanning four months.

The panels, labeled A to H, each focus on a different climatological variable. The bars in these panels depict the correlation coefficients for each municipality. Panel A, examining total monthly rainfall, shows that most municipalities exhibit no significant correlation, marked by red bars. However, a few municipalities show a negative trend, shown by blue bars, with Cimitarra uniquely displaying both significant positive and negative correlations. Panel B, analyzing average monthly temperature, reveals predominantly insignificant results, following the pattern seen in Panel A.

Panels C and D investigate the absolute maximum and minimum monthly temperatures, respectively. These panels show a combination of significant negative correlations and non-significant results, suggesting an inverse relationship between extreme temperatures and incidence rates. Panels E and F, which look at average maximum and minimum monthly temperatures, demonstrate both significant positive and negative correlations, indicating variable effects of average temperatures on incidence rates across municipalities.

Panels G and H focus on the differences in average and maximum temperatures, predominantly showing non-significant correlations. This suggests that temperature variability has a lesser impact on incidence rates compared to average or extreme temperatures.

#### 3.2.2. Precipitation

Statistically significant positive correlations: San José del Guaviare was the only municipality to show a positive correlation with statistical significance, ranging from weak to moderate; 0.348 (*p* 0.044, CI 0.0–0.62), 0.485 (*p* 0.004, CI 0.15–0.72), and 0.469 (*p* 0.007, CI 0.12–0.71) for 2, 3, and 4 lagged months, respectively. With a one-month lag, the coefficient was positive but not significant (Figure 1A).

Statistically significant negative correlations: Anorí, Florencia, La Macarena, Rioblanco, Rovira, Samaná, Santa Rosa del Sur, and San Vicente del Caguán showed at least one negative correlation with statistical significance at one of the four lag times. The strongest correlations were Santa Rosa del Sur in the 2nd month (Rho −0.629 *p* < 0.001, CI −0.81 to −0.34) and Rovira in the 2nd and 3rd month (Rho −0.463 *p* 0.006, CI −0.70 to −0.13 and Rho −0.552 *p* 0.001, CI −0.76 to −0.23, respectively). The rest of the correlation coefficients for these eight municipalities ranged between −0.4 and −0.15. The 2-month lag time showed the highest number of statistically significant correlations. None of these eight municipalities showed statistically significant positive correlations.

Cimitarra was the only municipality that showed divergence in the direction of the correlation coefficient. It showed a significant weak negative correlation for the first month (Rho −0.348 *p* 0.041, CI −0.62 to −0.01) and a significant weak positive correlation for the fourth month (Rho 0.37 *p* 0.037, CI 0.01 to 0.64) (Figure 2).

Chaparral, El Carmen de Chucurí, Tumaco, El Carmen de Bolívar, and Valdivia were the only municipalities that did not show statistical significance at any lag times for the total monthly rainfall variable.

#### 3.2.3. Temperature

The variable with the highest number of significant correlations was Average monthly temperature [5], followed by Average monthly maximum temperature [4], Average temperature difference [4], Average minimum monthly temperature [3], Maximum temperature difference [3], Absolute maximum monthly temperature [2], and Absolute minimum monthly temperature [2] (Table 3).

San José del Guaviare was particularly remarkable, showcasing a high number of significant correlations across multiple variables. This municipality registered three significant correlations for Total Monthly Rainfall (TMR), Average Monthly Temperature (T_AVER), Average Maximum Monthly Temperature (T_AVER_MAX), Maximum Temperature Difference (T_DIF), and Average Temperature Difference (T_AVER_DIF). This pattern underscores its sensitivity to climatic changes. Tumaco also demonstrated considerable variability, with two significant correlations for Average Monthly Temperature and one each for Absolute Minimum Monthly Temperature, Average Maximum Monthly Temperature, and Average Minimum Monthly Temperature.

Cimitarra and Rovira were notable for exhibiting significant correlations across more than one variable, with Cimitarra showing two for TMR and one for Absolute Maximum Monthly Temperature and Rovira showing two significant correlations for TMR. Contrastingly, several other municipalities, including El Carmen de Chucurí, El Carmen de Bolívar, Valdivia, Samaná, Santa Rosa del Sur, Rioblanco, Anorí, San Vicente del Caguán, and Florencia, had entries marked with “-”, indicating either no available data or no significant correlations found for most of the variables. Chaparral displayed a distribution across various temperature-related variables, although each only showed one significant correlation.

In most of the municipalities in which statistical significance was identified, the correlations turned out to be negative, mostly with coefficients of weak to moderate strength (San José del Guaviare Rho −0.509 *p*-value 0.002, CI −0.73 to −0.19 with Average monthly temperature being one of the strongest). Only the Chaparral showed a positive correlation with the Average temperature difference (Figure 3).

### 3.3. Locally Weighted Scatterplot Smoothing

The scatter plots shown in Figure 4 illustrate the relationship between two environmental variables, total monthly rainfall and average monthly temperature, and the incidence of a particular condition or event, with a 2-month lag. The first plot examines total monthly rainfall against incidence rates, displaying a broad scatter of data points with a relatively flat LOWESS line, indicating that rainfall within the observed range does not correspond with significant changes in incidence rates. This lack of a strong linear relationship suggests that rainfall alone may not significantly influence incidence rates, reflecting the mixed results from previous correlation analyses, which showed both positive and negative correlations across different municipalities.

The second plot focuses on average monthly temperature, revealing a more pronounced pattern. Here, incidence rates increase with rising temperatures from 24 °C up to approximately 26 °C, after which they begin to decline, suggesting a parabolic relationship. This pattern indicates that there is an optimal temperature range that correlates with higher incidence rates, and temperatures beyond this range tend to reduce incidence. The application of LOWESS fitting is particularly important for these variables due to their high number of statistically significant correlations, which helps visualize this relationship clearly.

The 2-month lag on the incidence rates corresponds to the mean incubation period of the disease as outlined by the WHO, which is crucial for understanding the time-delayed effects of environmental variables on disease transmission dynamics. This analysis underscores the complexity of environmental influences on disease transmission, highlighting temperature sensitivity where certain temperatures may facilitate the breeding of vectors or survival of pathogens, thereby increasing transmission up to a point before becoming detrimental. Conversely, the minimal impact of rainfall suggests that other unmeasured environmental or socio-economic factors might play more significant roles in disease transmission.

## 4. Discussion

### 4.1. Main Findings

The study covered 6141 cases of cutaneous leishmaniasis reported in 15 Colombian municipalities between 2017 and 2019, constituting 32.2% of the national total. Most affected were males, especially those aged 20–29, who comprised nearly 44% of cases. The majority of cases, 86.9%, were reported in rural areas, with significant representation from military personnel and farmers, who are often exposed to environments conducive to transmission. Over 65% of the cases involved economically vulnerable populations under subsidized social security.

The research also analyzed the distribution of cases in various geographical and altitudinal contexts, identifying high incidences in San José del Guaviare and Tumaco, which are at lower altitudes and potentially more susceptible due to climatic conditions. Statistical analysis revealed complex relationships between climatic factors and leishmaniasis incidence. Temperature showed significant correlations in some municipalities, with both positive and negative impacts on leishmaniasis incidence observed, depending on the specific climatic variable and municipality.

San José del Guaviare stood out for its high number of significant correlations across multiple climatic variables, indicating a particular sensitivity to climatic fluctuations. On the other hand, rainfall showed a less significant correlation with leishmaniasis incidence, suggesting that while climatic factors are influential, their effects are nuanced and potentially mediated by other environmental or socioeconomic factors.

### 4.2. Environmental Influences on Disease Transmission

Leishmaniasis is recognized as a climate-sensitive disease. The primary goal of this study was to discern the relationship between certain climatic variables and the incidence of CL in municipalities known for annually reporting the highest number of cases. The analysis reveals dynamic responses in CL incidence due to fluctuations in rainfall and temperature across different eco-epidemiological settings. Studies across endemic countries, both in the Old and New Worlds [21,23,31,39,40,41], have investigated the influence of temperature, precipitation, humidity, and other environmental factors on the incidence of CL, showing diverse outcomes across geographic regions. In particular, the relationship between plant density, vector density, and climatic factors such as rainfall has been highlighted in various studies.

Toumi et al., 2012 [42] consider that the increase in the density of plants that serve as food for certain reservoir rodents is associated with an increase in vector density, which affects disease transmission. In Brazil, a study found that higher levels of rainfall and relative humidity are related to an increase in vector density, probably due to favorable changes in the microhabitat of sandflies and reservoirs [43]. Additionally, an ecological niche model revealed that *Lutzomyia* (*N*.) *whitmani* prefers an intermediate vegetation density index (NDVI) associated with moderate rainfall when evaluating the environmental suitability of this species and the incidence of CL in Brazil [44]. Particularly, in the Andean region of Colombia, a positive association was found between CL incidence and tropical rainforest coverage and a negative association with livestock agroecosystems. This is probably due to the destruction of the natural habitat of vectors and reservoirs by replacing the forest with crops and pastures [24].

It is evident that localized ecological characteristics significantly influence the disease’s prevalence and spread. The heterogeneity in outcomes across various studies highlights the complexity of predicting disease trends based solely on broad environmental parameters. For instance, regions experiencing high-temperature variability may see peaks in CL incidence during warmer months when vectors are more active and human exposure increases, often due to agricultural or recreational activities.

Additionally, wind speed plays a crucial role, as there is a negative association between vector abundance and wind speed; as wind speed increases, the capture of sandflies decreases [45,46,47], which may be related to the fragility of sandflies that limits their movement, thus impacting disease transmission.

Quintana et al., 2020 [48] elucidated the complex interactions between various environmental factors and the abundance of CL vectors, highlighting the influence of both macro and micro-scale environmental conditions on vector populations and the transmission dynamics of leishmaniasis. Key findings include the positive correlation between tree cover and the increased abundance of *Nysomyia whitmani*, indicating that vegetation provides critical shelter and breeding grounds. Urban services such as garbage collection and the presence of blood sources like chickens and dogs significantly correlate with the abundance of *Lutzomyia longipalpis*, underscoring the impact of urban infrastructure on vector habitats. Additionally, moisture levels, as indicated by NDWI, have a varied impact on vector species, with higher moisture levels linked to reduced presence of *L. longipalpis*. The study also pointed to the importance of microclimatic variations, human activity, and domestic animals in shaping vector ecology. These insights stress the complexity of predicting vector-borne disease dynamics based on environmental parameters alone and highlight the necessity for localized studies to tailor vector control strategies effectively [48].

Haider et al., 2017 conducted a comprehensive study in Denmark that uncovered significant discrepancies between meteorological temperatures reported by standard weather stations and the actual microclimatic temperatures observed at various heights in specific habitats, with variations up to 5 °C. These findings highlight the inadequacy of relying solely on broad meteorological data for disease outbreak predictions. The study demonstrated that even minor changes in local temperature and humidity can significantly affect vector behavior and disease transmission. Such localized climate data are crucial for accurately predicting disease outbreaks and crafting targeted interventions, illustrating how areas with favorable microclimatic conditions, like dense vegetation and moderate humidity, can support higher vector densities, whereas less ideal conditions can diminish vector populations. This nuanced understanding of microclimatic effects is vital for formulating precise public health strategies to effectively manage vector-borne diseases such as leishmaniasis.

Precipitation patterns also play a dual role in influencing vector populations and disease transmission. On one hand, adequate rainfall can create ideal breeding conditions for sand flies by providing the necessary moisture for larval development. On the other hand, heavy rainfall can flood breeding sites and disrupt the lifecycle of the vector, thereby reducing transmission rates. This complex interplay indicates that local water management practices, such as irrigation or the construction of water reservoirs, could inadvertently affect the prevalence of leishmaniasis by altering habitat suitability for vectors [49,50].

The analysis of the research by Haider et al., 2017 [49] and the additional insights provided by Abiodun et al., 2016 [51] and Koch et al. [50] reveal critical interactions between precipitation patterns and vector-borne diseases like leishmaniasis. Haider et al. [49] underscored the critical role of localized climate data over generalized meteorological statistics in predicting disease outbreaks due to the discrepancies in microclimatic temperatures. This is essential because small variations in microclimate can significantly affect vector behavior and disease transmission dynamics.

From Abiodun et al. [51], it becomes apparent that not only temperature but also rainfall significantly impacts the population dynamics of vectors. Their findings highlight that while adequate rainfall provides necessary conditions for larval breeding, excessive rainfall can flush out breeding sites, thereby disrupting the lifecycle of vectors and potentially reducing disease transmission. This dual impact of precipitation is consistent with the observations made by Haider et al. regarding the complex interplay of climatic factors.

Koch et al. [50] extend this understanding by modeling the future distribution changes of leishmaniasis vectors under varying climate scenarios. They project a northward expansion of climatically suitable areas for these vectors in Europe due to climate change. This suggests that regions experiencing increased rainfall and warmer temperatures might see a heightened risk of leishmaniasis, assuming vector species follow these climatic shifts. Their study further emphasizes the importance of precipitation, alongside temperature, in influencing vector habitats and disease risks.

With the ongoing changes in global climate, long-term trends in temperature and precipitation are expected to impact leishmaniasis transmission globally. Regions that currently experience lower rates of leishmaniasis could become more susceptible if global warming leads to more favorable conditions for sand flies [52]. Modeling studies incorporating long-term climatic projections are crucial for anticipating future changes in disease patterns and preparing for potential shifts in disease hotspots. Rupasinghe et al., 2022 [52] characterize global warming as a rapid increase in Earth’s average surface temperature over the past century, predominantly driven by anthropogenic greenhouse gas emissions. This warming has induced substantial geoclimatic variations, such as changes in land and ocean temperatures, precipitation patterns, and the frequency of extreme weather events like heavy rains and floods. Such climatic shifts are essential for understanding disease dynamics, as they can significantly alter the habitats and behaviors of vector populations, including sand flies—vectors known for transmitting leishmaniasis. Historically confined to southern Europe below latitude 45° N and altitudes under 800 m, sand flies have extended their range northward to latitude 49° N, influenced by rising temperatures and increased precipitation in Northern Europe. This expansion has made regions such as Germany, Austria, Switzerland, and along the Atlantic coast newly suitable for sand fly habitation, potentially leading to the establishment of new endemic foci of leishmaniasis in Central and Northern Europe, areas that previously only reported imported cases. On the other hand, excessively hot and dry conditions might reduce sand fly populations in southern latitudes. Predictions suggest a potential northward shift of sand fly populations reaching as far as Great Britain and Scandinavia by 2061–2080. Moreover, in Brazil, changing conditions are expected to significantly increase leishmaniasis cases, with a similar trend projected from Mexico to the southern United States, potentially doubling the number of individuals exposed to the disease by 2080.

Curtin and Aronson, 2021 [53] focuses on leishmaniasis in the United States, a region of historically low endemicity. The researchers discuss the potential for increased disease transmission due to climate change, including the emergence of competent vectors and the mobility of the population. For instance, there is a concern about how global warming and other geoclimatic variations could expand the geographical range of vectors, increasing the risk of disease in previously non-endemic areas.

To manage leishmaniasis effectively, an integrative approach is essential, combining disease surveillance, vector control, environmental management, and public health education. Such strategies should be adaptable to local conditions and responsive to changes in environmental factors that influence disease transmission. Public health authorities must collaborate closely with environmental and agricultural agencies to implement landscape management practices that reduce vector habitats while promoting public awareness about preventive measures [52,53]. Moreover, this approach must be framed within the “One Health” concept, which recognizes the interconnection between human health, animal health, and the environment. By adopting a One Health approach, it is possible to address leishmaniasis not only from a human perspective but also considering animal health and the ecosystems that support the vectors and reservoirs of the disease. This requires multidisciplinary work that includes veterinarians, ecologists, public health experts, and local communities, all working together to design and implement strategies that consider all dimensions of ecosystem health. This integrated approach is crucial for developing more effective and sustainable interventions that can reduce the incidence of leishmaniasis in the long term [54,55].

### 4.3. Sociodemographic Impacts and Cultural Factors

The sociodemographic analysis of the cases highlighted a higher prevalence among males, middle-aged individuals, those employed in outdoor activities, particularly in wooded/jungle settings, and populations with lower resources and higher socioeconomic vulnerability (Table 2).

CL is prevalent across the main regions of the country [27,56], with most of the included localities situated in the Andean region (Figure 2 and Figure 3). According to the Köppen climate classification system, the Andean region of the country is characterized by a wet equatorial climate, tropical wet-dry climate, and temperate zones [57,58]; the remaining municipalities analyzed are characterized by tropical rainy climate zones as well. The contrast between these climatic regions and those of other geographical areas, such as Iran—one of the six most endemic countries for CL [59], predominantly featuring arid and semi-arid climates—illustrates the heterogeneous behavior of different vector species, reservoirs and parasites involved in the disease and their varied responses to climatic variability. The impact of climatic variability seems to differ according to geography and the specific species of vectors and parasites involved [21,60,61].

In the analysis of temperature, the monthly mean temperature exhibited the highest number of significant correlations, displaying negative correlations of low to moderate intensity with CL incidence in Tumaco and San José del Guaviare, albeit without statistical significance for other municipalities. The average monthly maximum temperature also showed moderate negative correlations in San José del Guaviare and Tumaco. A few significant relationships between temperature variables and CL incidence were found in the remaining municipalities, mostly negative, which aligned with findings from other studies [21,22,24,60,62,63,64,65]. This corresponds with the finding of increased longevity of certain Phlebotomine sand fly species in Colombia with temperature reduction, thereby enhancing their vectorial capacity by increasing encounter rates [66].

Moreover, the global analysis depicted the relationship between monthly mean temperature and CL incidence with a 2-month lag for the five municipalities. Results indicate a specific temperature range, approximately between 24 °C and 28 °C, that relates with increased number of cases, as observed for other vector-borne diseases [67].

Climate impacts CL dynamics through its effect on vector population density and ecological behavior; meteorological parameters influence vector reproduction and behavior [68], with vector abundance linked to CL cases [12]. Vectors show sensitivity to abrupt temperature changes, suggesting an optimal development temperature range identified in experimental studies across various sandfly and Leishmania species [16,69], which respond differently to environmental stimuli.

Humidity plays a crucial role in vector development and reproduction [40,70]. The correlation between total monthly rainfall and CL incidence divided municipalities into three categories: those with a positive correlation, those with a negative correlation, and those with no significant correlation found. San José del Guaviare exhibited the highest number of significant correlations with positive, low, and moderate intensity; other municipalities with significant correlations demonstrated an inverse relationship between rainfall and CL incidence. Municipalities such as San José del Guaviare, La Macarena, and San Vicente del Caguán exhibited different responses to similar climatic stimuli despite geographical proximity. Given the large area of these municipalities, the weather stations, located in specific points, might not fully capture the true climate behavior across the entire geographic division, potentially explaining the divergent results.

Most locations showed an increase in CL cases with decreased rainfall, which corresponds with other authors’ findings in the region [26,31]. This may suggest that excessive rainfall impedes vector flight and floods may hinder larval survival [29,50,71], but it may also hamper access to sylvatic areas, limiting occupational exposure. However, the overall LOWESS analysis exhibited stable incidence across different monthly rainfall ranges. While some studies report positive correlations between rainfall and case numbers [24], others have not established a link between precipitation and CL incidence [39,63].

The heterogeneous results in correlation analysis and the apparent lack of influence of rainfall on the global analysis, contrasted with evidence of conflicting relationships between disease incidence and rainfall [24,39,41,60,62], suggests that in regions where a minimal necessary humidity is maintained, monthly precipitation does not significantly affect CL incidence unless it is excessive which disrupts vector ecological dynamics and limits occupational exposure due to behavioral changes.

The works by Okwor et al. [9], Wijerathna et al. [10] and Valero et al. [11] discuss the substantial economic and psychological burdens of leishmaniasis on affected populations. The disease predominantly impacts economically active individuals, often primary income earners, exacerbating the financial strain due to medical costs and income loss during treatment. For instance, Wijerathna et al. [10] highlights the substantial costs associated with treatment in Sri Lanka, including both direct (travel and treatment) and indirect (loss of income) expenses. Okwor et al. [9] stress the need for vaccine development given the disease’s socioeconomic impact, particularly in rural areas where medical access is limited.

The insights from these studies suggest that an integrated approach involving socioeconomic support, environmental management, and robust health systems is crucial for effective leishmaniasis management. Sunyoto [7] and colleagues discuss the necessity of free diagnosis and treatment to prevent catastrophic health expenditures, highlighting the importance of accessible healthcare in managing the disease. Furthermore, the review by Valero et al. [11] underscores the need for consistent access to treatments and the development of new diagnostic tools, especially in rural areas where options are limited.

### 4.4. Study Limitations and Future Research

Some limitations of this study are the lack of inclusion of other variables (such as socioeconomic or additional environmental factors) that could affect the climate–CL relationship, the potential underreporting of cases due to healthcare access difficulties in remote areas, and the limited climatological data availability which resulted in the exclusion of many municipalities of the temperature analysis. While correlation analysis was useful for identifying non-causal relationships, designing a study that considers confounding factors and narrows the geographic area of study could yield more precise insights into the climatic–epidemiology relationship.

In addition to the limitations, this study’s methodological approach reveals several areas that could be enhanced in future research. For instance, integrating data on land use changes, population movements, and economic activities might offer a clearer picture of the socioenvironmental dynamics affecting disease spread [11].

Future studies could benefit from a more refined geographic focus, perhaps concentrating on a smaller number of municipalities but with more detailed data collection, including more frequent and widespread climatological measurements. It is also advisable to improve the reporting systems for leishmaniasis cases to mitigate the issue of underreporting, especially in remote and underserved areas. Utilizing mobile health technologies or community-based reporting could enhance data accuracy and timeliness [72,73].

Moreover, associating these findings with the global impact of climate change and adopting a One Health approach could be necessary. Recognizing the interconnectedness of human, animal, and environmental health allows for a more holistic understanding and management of zoonotic diseases like leishmaniasis. It aligns with global trends toward integrated health strategies that consider the impacts of global environmental changes on disease patterns [74,75,76].

Implementing these suggestions could not only refine the understanding of the climatic–epidemiology relationship but also align with global health priorities that emphasize sustainable, cross-sectoral strategies to manage and prevent zoonotic diseases effectively. This approach is particularly relevant as climate change continues to alter disease dynamics worldwide, highlighting the need for resilient health systems that can adapt to these changes while considering the broader ecological and social determinants of health.

## 5. Conclusions

The findings suggest that variations in temperature and rainfall may influence the behavior of the disease, even in areas with non-seasonal climates such as Colombia. The impact on the disease is likely indirect, primarily affecting the ecology of the vector and the parasite. The data reveals a significant correlation between current rainfall and temperature with the subsequent incidence of CL; however, this relationship is not linear, as these are not the sole factors involved. Instead, a multitude of elements alter the dynamics of the vector, parasite, reservoirs, and human behavior. Furthermore, the nature of the climate–incidence relationship is found to be highly variable across different geographical regions.

Enhancing our understanding of these interactions is crucial in a world experiencing climate change. Future studies should aim to examine the climate–CL relationship within more narrowly defined geographical areas to improve the precision of the data. A deeper insight into the determinants of this relationship will enable the development of localized predictive models and the formulation of targeted strategies to mitigate the impact of this condition.

## Figures and Tables

**Figure 1 pathogens-13-00462-f001:**
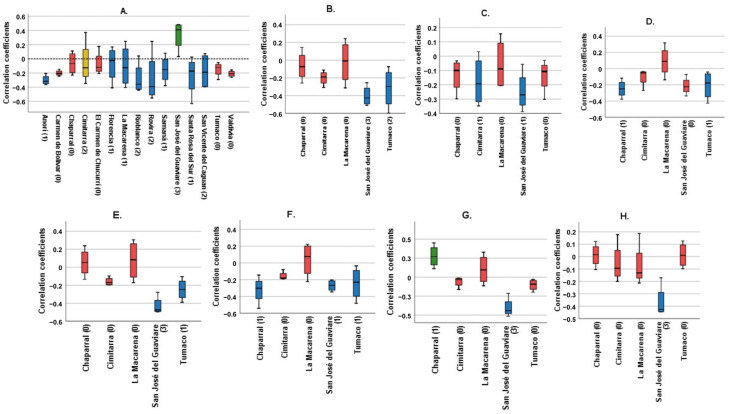
Correlation coefficients of the different climatological variables for each municipality. Blue: Municipalities with statistically significant correlations with a negative trend. Green: Municipalities with statistically significant correlation with a positive trend. Red: Municipalities without statistically significant correlations. Yellow: Cimitarra was the only locality to present a significant positive correlation and a significant negative correlation for the same variable. (**A**) Total monthly Rainfall, (**B**) Average monthly temperature, (**C**) Absolute maximum monthly temperature, (**D**) Absolute minimum monthly temperature, (**E**) Average maximum monthly temperature, (**F**) Average minimum monthly temperature, (**G**) Average temperature difference, (**H**) Maximum temperature difference. In parentheses is the number of significant correlations for each municipality. Source: Own elaboration.

**Figure 2 pathogens-13-00462-f002:**
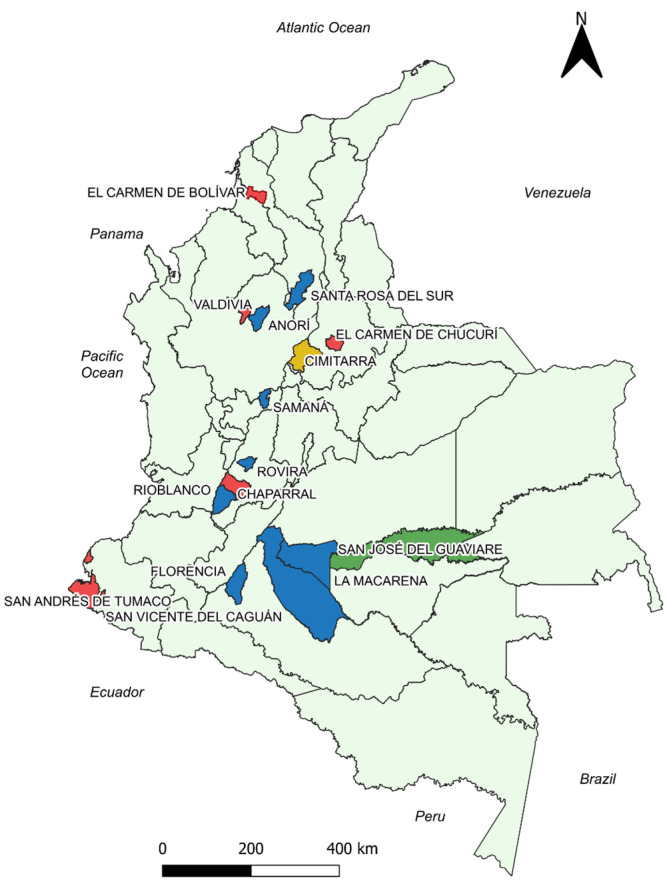
Georeferenced map of the municipalities with the results of the correlation analysis between the cumulative incidence variable and the total monthly precipitation. Blue: Municipalities with statistically significant negative correlation. Green: Municipalities with statistically significant positive correlation. Red: Municipalities without statistically significant correlations. Yellow: Cimitarra presented a weak positive and a negative correlation, but its tendencies were to be negative. Source: Own elaboration.

**Figure 3 pathogens-13-00462-f003:**
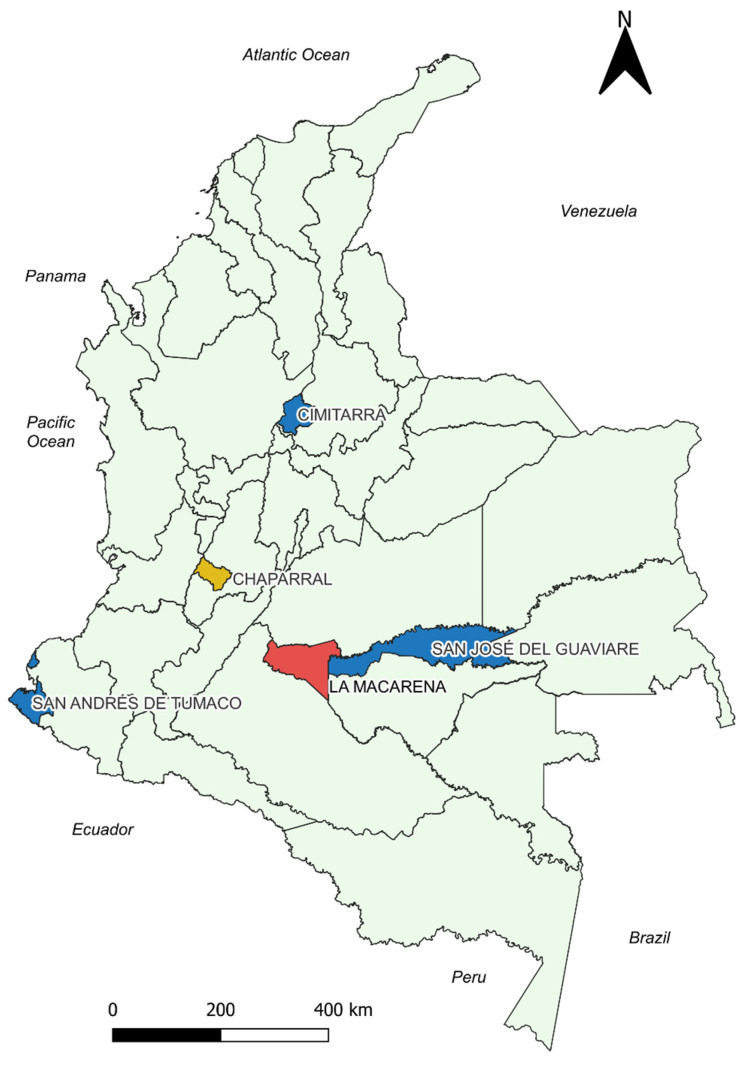
Georeferenced map of the municipalities with the results of the correlation analysis between the cumulative incidence variable and temperature. Blue: Municipalities with statistically significant negative correlation. Red: Municipalities without statistically significant correlations. Yellow: Chaparral presented positive correlation for T_AVER_DIF and negative correlations for T_ABS_MIN, T_AVER_MIN. Source: Own elaboration.

**Figure 4 pathogens-13-00462-f004:**
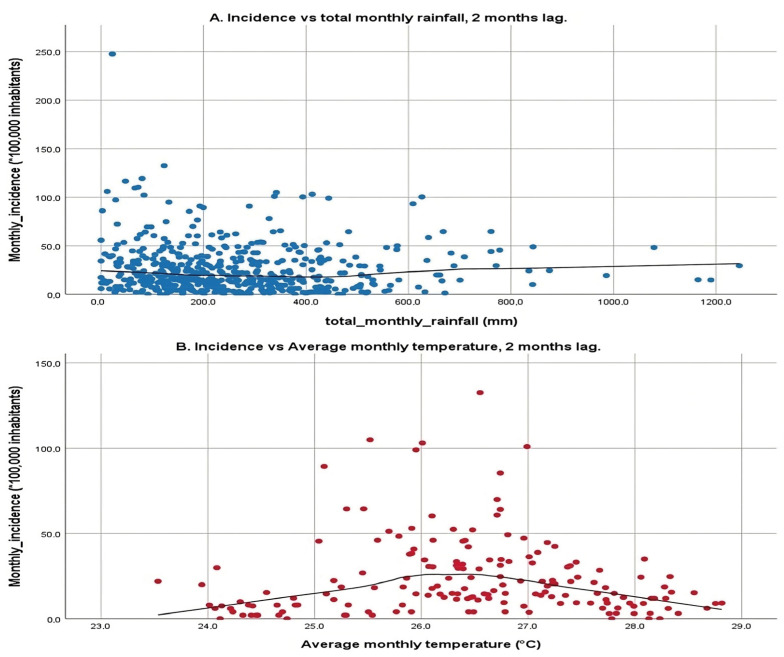
Simple scatter plot with line fitting according to LOWESS with a 2-month lag. (**A**): X-axis: total monthly rainfall, Y-axis: Incidence with a 2-month lag. (**B**): X-axis: average monthly temperature, Y-axis: Incidence with a 2-month lag. Scatter plot of the climate variable vs. the cumulative incidence at monthly intervals with a 2-month lag for each municipality over the 3 years (January 2017 to December 2019).

**Table 1 pathogens-13-00462-t001:** Summary of the sociodemographic characteristics of the population included in the study.

Variable	Genre		Total N (%)
	Female	Male	
Age			
0–9	276	282	558 (9.09)
10–19	277	862	1139 (18.55)
20–29	207	2491	2698 (43.93)
30–39	164	704	868 (14.13)
40–49	120	267	387 (6.3)
50–59	82	208	290 (4.72)
60–69	37	84	121 (1.97)
70–79	19	44	63 (1.03)
80–89	5	11	16 (0.26)
>90	1	0	1 (0.02)
Year			
2017	516	1698	2214 (36)
2018	320	1736	2056 (33.5)
2019	352	1519	1871 (30.5)
Area of Presentation of the Case			
Town	59	300	359 (5.8)
Village	88	360	448 (7.3)
Rural	1041	4293	5334 (86.9)
Occupation			
Military	3	2576	2579 (41.99)
Farmer/Forestal worker	43	1211	1254 (20.42)
Student/Teacher	359	557	916 (14.91)
Housekeeper	551	18	569 (9.26)
Unemployed/ND ^1^/Various	195	325	520 (8.46)
Mining/Construction worker	2	168	170 (2.76)
Other	35	98	133 (2.16)
Social Security Regime			
Contributive	72	755	827 (13.46)
Special	2	267	269 (4.38)
Indeterminate	16	32	48 (0.78)
No insurance	29	105	134 (2.18)
Exception (Military/Teachers)	2	866	868 (14.13)
Subsidized	1067	2928	3995 (65.05)
Total N (%)	1188 (19.3%)	4953 (80.7%)	6141 (100%)

^1^ ND: No data.

**Table 2 pathogens-13-00462-t002:** The distribution of cases by department and municipality and their classification according to altitude in meters above sea level (M.A.S.L. ^1^) for the 15 municipalities during the years 2017 to 2019.

Region	Municipality	Altitud (M.A.S.L ^1^)	Number of Cases per Municipality	Number of Cases per Region
Antioquia	Anorí	1550	270	517
	Valdivia	1064	247	
Bolívar	El Carmen de Bolívar	155	382	675
	Santa Rosa del Sur	600	293	
Caldas	Samaná	1250	253	253
Caquetá	San Vicente del Caguán	250	190	342
	Florencia	266	152	
Guaviare	San José del Guaviare	189	1018	1018
Meta	La Macarena	233	234	234
Nariño	Tumaco	3	1934	1934
Santander	El Carmen de Chucurí	500	344	476
	Cimitarra	159	132	

^1^ M.A.S.L.: Meters above sea level. Source: own elaboration, Data provided by SIVIGILA, Instituto Geográfico Agustín Codazzi. (https://www.colombiaenmapas.gov.co/#, accessed on 18 September 2023, Bogotá, Colombia).

**Table 3 pathogens-13-00462-t003:** Number of statistically significant correlations for the different climatological variables1 in the 15 municipalities during the years 2017 to 2019.

Municipality	TMR	T_AVER	T_ABS_MAX	T_ABS_MIN	T_AVER_MAX	T_AVER_MIN	T_DIF	T_AVER_DIF
Tumaco	0	2	0	1	1	1	0	0
San José del Guaviare	3	3	1	0	3	1	3	3
La Macarena	1	0	0	0	0	0	0	0
Chaparral	0	0	0	1	0	1	0	1
Cimitarra	2	0	1	0	0	0	0	0
Rovira	2	-	-	-	-	-	-	-
El Carmen de Chucurí	0	-	-	-	-	-	-	-
El Carmen de Bolívar	0	-	-	-	-	-	-	-
Valdivia	0	-	-	-	-	-	-	-
Samaná	1	-	-	-	-	-	-	-
Santa Rosa del Sur	1	-	-	-	-	-	-	-
Rioblanco	2	-	-	-	-	-	-	-
Anorí	1	-	-	-	-	-	-	-
San Vicente del Caguán	2	-	-	-	-	-	-	-
Florencia	1	-	-	-	-	-	-	-

TMR: Total monthly rainfall, T_AVER: average monthly temperature, T_ABS_MAX: absolute maximum monthly temperature, T_ABS_MIN: absolute minimum monthly temperature, T_AVER_MAX: average maximum monthly temperature, T_AVER_MIN: average minimum monthly temperature, T_DIF: maximum temperature difference, and T_AVER_DIF: average temperature difference. -: No data.

## Data Availability

The original contributions presented in the study are included in the article, further inquiries can be directed to the corresponding authors.

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
