# Peer review of "The Impact of Climatological Factors on the Incidence of Cutaneous Leishmaniasis (CL) in Colombian Municipalities from 2017 to 2019"

_pathogens, 2024, doi:10.3390/pathogens13060462_

Round 1

Reviewer 1 Report

Comments and Suggestions for Authors

-Add information on the relationship of vegetation and wind density on their effect on vector density.

-You can add information about prevalence of leishmaniasis cases

-Do you probe the normalized curve of numeric variables? In order to use parametric or not paremetric statistic.

Author Response

We sincerely thank Reviewerfor their valuable and insightful comments, which have significantly improved our manuscript. Your suggestions have been addressed and incorporated into the document as follows:

-Add information on the relationship of vegetation and wind density on their effect on vector density.

Response: It was added in the document.

-You can add information about prevalence of leishmaniasis cases

Response: It was added in the document.

-Do you probe the normalized curve of numeric variables? In order to use parametric or not paremetric statistic.

Response: To analyze the relationship between climatological variables (precipitation and temperature) and the accumulated incidence of CL, correlation analysis was conducted. Initially, a hypothesis test was performed to determine the statistical distribution of the data using the Shapiro-Wilk test, which indicated that the data had a non-normal statistical distribution, and therefore, Spearman's correlation was chosen. The Shapiro-Wilk hypothesis test (chosen because it involved 36 data points, one for each month analyzed over the 3 years) revealed that the climatological data, especially the incidences, had a non-normal statistical distribution. This analysis was conducted for each municipality, and the results were similar, leading to the choice of non-parametric statistics.

Reviewer 2 Report

Comments and Suggestions for Authors

This study aims to determine the impact of temperature and rainfall fluctuations on the incidence of cutaneous leishmaniasis (CL), a vector-borne disease, performing an analysis of 6141 reported cases in Colombian 347 municipalities between 2017 and 2019. The analysis revealed both significant positive and negative correlations, depending on locality and climate variables. The findings underscore the significant yet complex influence of climatic factors on CL incidence. The findings provided will enable to aid public health efforts by improving predictive models and crafting targeted interventions to mitigate the CL impact, particularly in regions vulnerable to climate variability.

Major comments:

The manuscript is well-structured but too long. I kindly recommend the Introduction and Discussion sections including conclusion section are better to be reduced extensively (about 2/3 volume), describing more concisely and comprehensively, focusing on the aim and main findings of the study; and also to be avoided unnecessarily citation/description of already published and/or widely known information.  

 Minor comments:

Line 32, “Leishmania”, should be italicized. All the scientific names should be in italic.  

Line 385, replace “N. whitmani” by Nysomyia whitmani.

Line 387, replace “L. longipalpis” by Lutzomyia longipalpis.

Line 506, replace “Phlebotomus species” by Phlebotomine sand fly species.

Comments on the Quality of English Language

Line 32, “Leishmania”, should be italicized. All the scientific names should be in italic.  

Line 385, replace “N. whitmani” by Nysomyia whitmani.

Line 387, replace “L. longipalpis” by Lutzomyia longipalpis.

Line 506, replace “Phlebotomus species” by Phlebotomine sand fly species.

Author Response

We sincerely thank Reviewerfor their valuable and insightful comments, which have significantly improved our manuscript. Your suggestions have been addressed and incorporated into the document as follows:

This study aims to determine the impact of temperature and rainfall fluctuations on the incidence of cutaneous leishmaniasis (CL), a vector-borne disease, performing an analysis of 6141 reported cases in Colombian 347 municipalities between 2017 and 2019. The analysis revealed both significant positive and negative correlations, depending on locality and climate variables. The findings underscore the significant yet complex influence of climatic factors on CL incidence. The findings provided will enable to aid public health efforts by improving predictive models and crafting targeted interventions to mitigate the CL impact, particularly in regions vulnerable to climate variability.

Major comments:

The manuscript is well-structured but too long. I kindly recommend the Introduction and Discussion sections including conclusion section are better to be reduced extensively (about 2/3 volume), describing more concisely and comprehensively, focusing on the aim and main findings of the study; and also to be avoided unnecessarily citation/description of already published and/or widely known information. 

Response: Changes were made to the introduction; however, we believe that the discussion, as it stands, is logically coherent and that the data are valuable because they support the interpretation of the results and the relevance of the study.

 Minor comments:

Line 32, “Leishmania”, should be italicized. All the scientific names should be in italic. 

Line 385, replace “N. whitmani” by Nysomyia whitmani.

Line 387, replace “L. longipalpis” by Lutzomyia longipalpis.

Line 506, replace “Phlebotomus species” by Phlebotomine sand fly species.

Comments on the Quality of English Language

Line 32, “Leishmania”, should be italicized. All the scientific names should be in italic. 

Line 385, replace “N. whitmani” by Nysomyia whitmani.

Line 387, replace “L. longipalpis” by Lutzomyia longipalpis.

Line 506, replace “Phlebotomus species” by Phlebotomine sand fly species.

Response:

All comments were changed.